

# Assessing the link between hygienic material use during menstruation and self-reported reproductive tract infections among women in India: a propensity score matching approach

Mahashweta Chakrabarty[1] and Aditya Singh[1,2]

[1] Department of Geography, Banaras Hindu University, Varanasi, Uttar Pradesh, India
[2] Girl Innovation, Research, and Learning (GIRL) Center, Population Council, New York, NY, USA

Corresponding author
Aditya Singh, adityasingh@bhu.ac.in

## ABSTRACT

**Background**. Reproductive tract infections (RTIs) present a substantial health concern for women, especially in developing nations such as India, where inadequate access to proper sanitation and hygiene facilities frequently results in suboptimal menstrual health and hygiene (MHH), exacerbating the risk of RTIs. In this study, we analysed the self-reported prevalence of RTIs among young women in India and evaluated the impact of hygienic menstrual material usage on these RTIs.

**Methods**. The study used information on 27,983 women aged 15–24 years, from the National Family Health Survey (NFHS-5) (2019-21). The prevalence of RTIs was calculated for all the states and UTs of India, and propensity score matching (PSM) technique was used to evaluate the impact of hygienic material use on RTIs among women in India.

**Results**. Every four out of 100 women reported RTIs in India in 2019–21. Notably, RTI prevalence displayed substantial state-level disparities. West Bengal exhibited the highest RTI prevalence at 9.3%, followed by Meghalaya, Arunachal Pradesh, and Himachal Pradesh, all surpassing 6%. In contrast, the lowest RTI rates were recorded in Puducherry at 0.9%, succeeded by Andaman and Nicobar Islands, Odisha, and Jammu & Kashmir, all registering rates below 2%. The PSM analysis revealed that women who utilized hygienic materials during menstruation exhibited a reduced prevalence of RTIs (referred to as the "treated group" with an Average Treatment Effect on the Treated (ATT) of 0.0315) compared to those who did not utilize such materials (referred to as the "control group" with an ATT of 0.0416).

**Conclusions**. The study underscores the critical significance of using hygienic materials during menstruation as a preventive measure against RTIs among women in India. The findings suggest the need for targeted interventions focused at promoting hygienic menstrual materials to reduce the prevalence of RTIs among women in India.

## INTRODUCTION

Reproductive tract infections (RTIs) have been identified as a significant global public health concern by the World Health Organization (WHO). This issue is particularly pronounced in developing countries, where it impacts a substantial proportion of reproductive-aged women, with an estimated prevalence ranging from 23% to 29% annually (*Pan American Health Organization, 2016*; *Dhabhai et al., 2022*). RTIs, stemming from various pathogens including bacteria, viruses, fungi, and parasites, affect both genders (*Patel et al., 2006*; *Bradford & Ravel, 2017*). However, the burden is disproportionately higher in women due to their unique anatomical and physiological characteristics, rendering them more susceptible to infections (*Klouman et al., 1997*; *Anbesu et al., 2023*).

Within low-middle-income countries, such as India, RTIs exert a significant impact on women's reproductive health (*Klouman et al., 1997*; *Baker et al., 2017*). They can lead to infertility, scarring and damage to reproductive organs, leading to blockages and impairments in the function of the fallopian tubes, and complications during pregnancy such as sepsis and postpartum haemorrhage (*Johnson et al., 2011*; *Racicot & Mor, 2017*; *Tsevat et al., 2017*; *Zeng et al., 2022*). In addition, women with RTIs are more vulnerable to HIV infection as these infections can cause genital inflammation and ulceration, providing an entry point for the virus (*Passmore et al., 2018*; *Mwatelah et al., 2019*). The symptoms associated with RTIs, such as abnormal discharge, itching, and pain, can inflict significant psychological distress on women, potentially resulting in stigmatization, shame, embarrassment, and social isolation (*London, 2005*; *Bilardi et al., 2013*; *Adolfsson et al., 2017*). Hence reducing the prevalence of RTIs is a crucial step in improving overall health and well-being, and these efforts can also promote gender equality and empower women to participate fully in social, economic, and political life, aligning with Sustainable Development Goals (SDGs)-3 and SDG-5 (*Feng et al., 2021*; *Sommer et al., 2021*; *Dhabhai et al., 2022*).

The maintenance of menstrual hygiene is a critical component of women's reproductive health (*Das et al., 2015*; *Wagh, Upadhye & Upadhye, 2018*; *Vishwakarma, Puri & Sharma, 2020*). Suboptimal menstrual health and hygiene (MHH), including inadequate menstrual product usage, lack of adequate cleansing practices, and the reuse of materials, have been found associated with an increased risk of RTIs, encompassing conditions such as bacterial vaginosis and urinary tract infections (*Sumpter & Torondel, 2013*). Some studies conducted in Bangladesh, Kenya, and Nepal have found that using hygienic materials such as sanitary pads, tampons, or menstrual cups can help prevent the growth of harmful bacteria and reduce the risk of RTIs (*Juyal, Kandpal & Semwal, 2014*; *Phillips-Howard et al., 2016*; *Torondel et al., 2018*; *Almeida-Velasco & Sivakami, 2019*; *Bhusal, 2020*; *Austrian et al., 2021*). Recent pieces of evidence in India suggest that women who used sanitary napkins had a lower risk of RTIs than those who used other materials like cloth, ash, or husk (*Torondel et al., 2018*; *MacRae et al., 2019*). One previous study also revealed that women who had access to education on MHH had a lower risk of RTIs than those who did not (*Pandit et al., 2014*). In addition, increased knowledge and awareness of hygiene

practices led to improved menstrual hygiene and reduced risk of infections among women in India (*Dube & Sharma, 2012*; *Malhotra et al., 2016*).

Most of the previous works examining the association between use of hygienic materials and RTIs among women are limited to small geographical areas and have been conducted in hospital settings (*Torondel et al., 2018*; *McCammon et al., 2020*; *Das et al., 2021*). Some cross-sectional studies, utilizing nationally representative survey data and employing regression analysis, have uncovered evidence suggesting an association between the usage of hygienic menstrual materials and the prevalence of RTIs in India (*Anand, Singh & Unisa, 2015*; *Vishwakarma, Puri & Sharma, 2020*). However, establishing a causal link between intervention and outcome in cross-sectional studies through regression analysis can be challenging due to potential selection bias. To address this, researchers often turn to quasi-experimental methods, such as propensity score matching (PSM), to mitigate selection bias (*Baser, 2007*; *Dixit, Dwivedi & Ram, 2013*). Notably, no prior studies have definitively established a causal link between hygienic material use and RTIs among Indian women using a quasi-experimental approach. This study aims to address this research gap.

Therefore, this article aims to assess the impact of hygienic material use during menstruation on the prevalence of RTIs among women in India. The study utilizes cross-sectional data from the National Family Health Survey-5 (NFHS-5) and employs PSM to establish a causal link between the two.

## DATA AND METHODS

### Data source

The present study used data from the NFHS-5, which was conducted between 2019 and 2021. NFHS-5, a nationally representative series of cross-sectional surveys, is designed to capture an array of vital information, including demographics, socioeconomics, maternal and child welfare, reproductive health, and family planning (*Ministry of Health and Family Welfare, Government of India, 2021*). Employing a two-stage stratified sampling method, a total of 664,972 households were selected for the survey, among which 636,699 were successfully interviewed, with a response rate of 98%. Additionally, within the households that were interviewed, a total of 747,176 women aged 15–49 were identified for individual interviews. Among them interviews were completed for 724,115 women aged 15–49, with a response rate of 97%. Among the total interviewed women, only 108,785 women were selected for the questions included in the state module. From those, ultimately 27,983 women aged 15–24 were chosen as a final sample for this study, as illustrated in Fig. 1.

### Ethics statement

This study relies on publicly available, de-identified dataset that does not contain any personally identifiable information about the survey participants. As such, there was no requirement for ethical permission. The NFHS-5 data utilized in this study is publicly accessible through the official website of Demographic and Health Surveys (DHS) at https://dhsprogram.com/data/available-datasets.cfm. It can be obtained by submitting a formal request to the DHS.

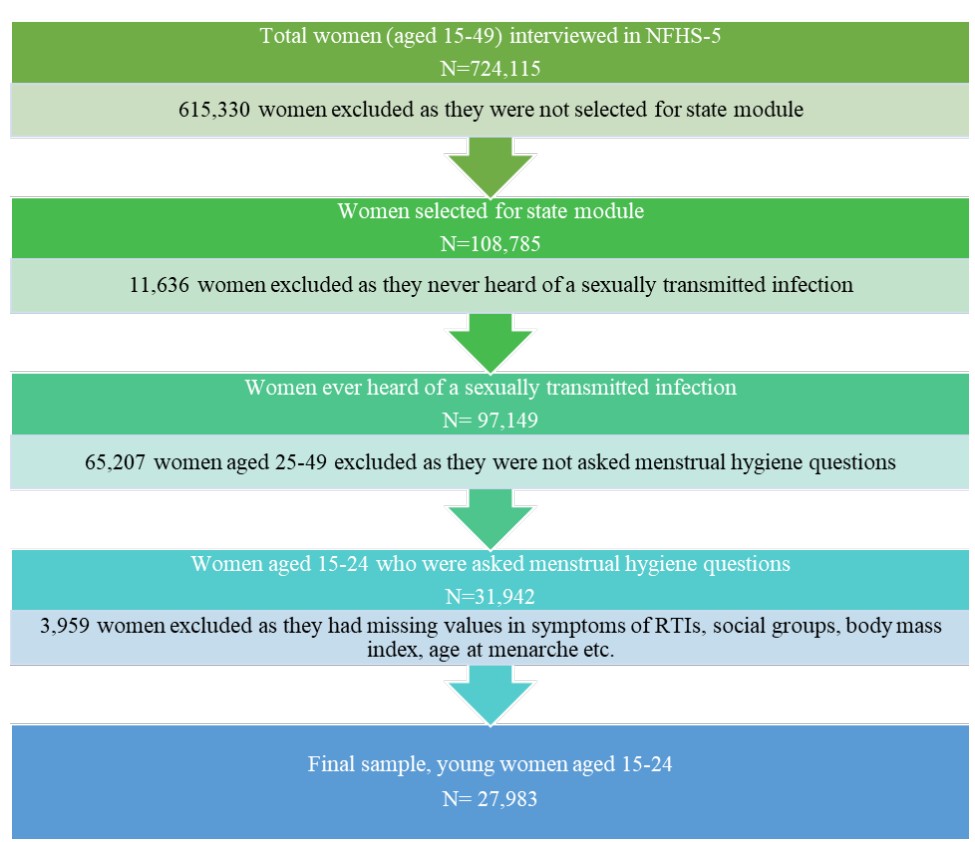

**Figure 1   Flow chart showing the process of selection of a sample from NFHS-5 dataset.**

## Outcome variable

In NFHS-5, women who were part of the state module and who reported being sexually active, regardless of their marital status, were asked two following questions:

"During the last 12 months, have you had a bad-smelling abnormal genital discharge?"

"During the last 12 months, have you had a genital sore or ulcer?"

From the responses obtained for the above questions, we created a binary outcome variable, with two categories: (a) 'women with RTIs'(coded as 1), *i.e.,* women who reported either or both of the symptoms mentioned above (abnormal genital discharge and/or genital sore or ulcer), (b) The remaining women were categorised as 'women without RTIs'(coded as 0).

## Treatment variable

In the NFHS-5 survey, women aged 15–24 were asked about the methods they used to prevent bloodstains from becoming evident during their menstrual periods. The responses were categorized into two groups:

*Hygienic Materials:* This category included sanitary napkins, locally made napkins, tampons, and menstrual cups.
*Unhygienic Materials:* This category included cloth, using nothing (implying no use of any protective material), and other materials not falling into the hygienic category.

Based on the responses, we created a binary outcome variable with two categories:

*Exclusive users of hygienic materials* (coded as "1"): These are women who exclusively utilized hygienic materials such as sanitary napkins, locally made napkins, tampons, and/or menstrual cups.

*Users of unhygienic materials* (coded as "0"): This category includes women who used cloth, other materials, nothing (indicating no use of any protective material), or used a combination of both hygienic and unhygienic materials (*Singh et al., 2022a*; *Singh et al., 2022b*).

Throughout this article, we have consistently used the terms 'use of hygienic materials' and 'exclusive use hygienic materials' interchangeably to convey the same concept. In our context, both terms refer to the practice of relying solely on hygienic materials, such as sanitary napkins, locally made napkins, tampons, and menstrual cups, for menstrual protection, without combining them with unhygienic materials or practices.

## Matching variables

In this study, a range of pertinent socioeconomic and demographic predictors identified in previous research on RTIs and MHM were selected as matching variables (*Ghebremichael, Paintsil & Larsen, 2009*; *Anand, Singh & Unisa, 2015*; *Brown, Gause & Northern, 2016*; *Torondel et al., 2018*; *Wagenaar et al., 2018*; *Vishwakarma, Puri & Sharma, 2020*; *Das et al., 2021*; *Cho & Yang, 2023*). These variables included: current age of respondents (in years), age at menarche (in years), years of schooling (in years), social groups (Scheduled Caste/Scheduled Tribe or SC/ST = 1, non-SC/ST = 0), religion (Hindu = 1, non-Hindu = 0), household wealth factor scores, exposure to mass media (exposure to at least one medium = 1, no mass media exposure = 0), discussion of menstrual hygiene (MH) with community health workers(CHW) (discussed = 1, not discussed = 0), working status (currently working = 1, not working = 0), bath during menstruation (take bath = 1, do not take bath = 0), consumption of alcohol (yes = 1, no = 0), place of residence (rural = 1, urban = 0), and region of residence (north, central, east, west, south, north-east).

## Statistical analysis

We begin by presenting a summary of the sample characteristics of the women (aged 15–24 years) included in our study. Subsequently, we calculate the prevalence of RTIs over the past year for these sampled women. We employed a Propensity Score Matching (PSM) approach to assess the impact of hygienic material use on RTI prevalence among women in India (*Rosenbaum & Rubin, 1985*; *Dimewiki, 2019*; *Kane et al., 2020*). For the analysis, we utilized the *psmatch2* command in Stata16 statistical software (*StataCorp, 2019*).

When randomized controlled trials, often regarded as the gold standard for assessing treatment effectiveness, are not feasible, and we are working with cross-sectional, observational, or non-experimental data, as in this study, quasi-experimental methods like PSM can serve as valuable tools for evaluating treatment effects (*Rosenbaum & Rubin, 1985*; *Dimewiki, 2019*; *Kane et al., 2020*). PSM replicates the randomization process

of randomized controlled trials, enabling robust causal inference, particularly when researchers have limited control over treatment assignment. By systematically matching individuals with similar probabilities of receiving treatment, PSM effectively addresses selection bias and creates comparable treatment and control groups.

In this study, we utilized the 1:1 nearest neighbor matching method with a caliper of 0.2 standard deviations of the logit of the propensity score. This approach was chosen based on recommendations from *Rosenbaum & Rubin (1985)* and *Austin (2011)* for its effectiveness in addressing imbalance and reducing bias by matching propensity scores between the treatment and control groups (*Rosenbaum & Rubin, 1985*; *Caliendo & Kopeinig, 2005*; *Austin, 2011*; *Greifer, 2022*).

Additionally, there are multiple reasons behind choosing 1:1 nearest neighbour matching over other matching methods (*e.g.*, k:1 nearest matching, kernel or local linear matching *etc.*). First, 1:1 nearest neighbour matching is relatively straightforward to implement. It involves pairing each treated unit with one control unit that is closest in terms of covariate values (*Greifer, 2022*). Second, it requires fewer computational resources compared to more complex matching methods, making it practical for large datasets such as NFHS, which is why several studies utilizing the NFHS dataset have consistently favoured nearest neighbour matching over other complex matching methods (*Dixit, Dwivedi & Ram, 2013*; *Dixit, Gupta & Dwivedi, 2018*; *Wilkinson, Mamas & Kontopantelis, 2022*). Third, this matching is more interpretable because each treated unit is directly matched to one control unit (*Stuart & Rubin, 2007*). This pairing can simplify the analysis and make it easier to explain to a non-technical audience. Furthermore, recent research by *Greifer (2022)* lends support to the preference for 1:1 nearest neighbor matching over k:1 matching, as an increase in k in the latter method can potentially exacerbate imbalance (*Greifer, 2022*). Also, nearest neighbor matching is better than kernel or local linear matching methods due to the latter occasionally including observations that are poor matches (*Caliendo & Kopeinig, 2005*).

The matching quality and balance were assessed by calculating the mean and median bias before and after matching using the *pstest* command in Stata 16. Sensitivity analysis was conducted using the Mantel-Haenszel bounds proposed by Becker to estimate the significance levels at different levels of hidden bias (*Becker & Caliendo, 2007*). The details of PSM, balancing test, and sensitivity analysis are provided in the appendix (see Appendix S1). For all analyses and estimations, the significance level was established at 0.05, corresponding to a confidence level of 95%.

## RESULTS

### Sample characteristics of the respondents

Table 1 represents the weighted percentage of women by background characteristics. The majority experienced their first menstruation between the ages of 13–15 years. About 47% of women were OBC, and most were Hindu. Only a few women had discussed MH with CHW in the past three months. About 16% of women reported no exposure to mass media. More than two-thirds of the women resided in rural areas. Half of the women reported

**Table 1  Percentage distribution of women by background characteristics, NFHS-5, 2019-21.**

| Background characteristics | Frequency ($N = 27,983$) | Weighted percentage |
|---|---|---|
| **Age (in years)** | | |
| 15–19 | 14,328 | 51.20 |
| 20–24 | 13,655 | 48.80 |
| **Age at menarche (in years)** | | |
| ≤12 | 4,640 | 16.58 |
| 13–15 | 22,193 | 79.31 |
| ≥16 | 1,150 | 4.11 |
| **Years of schooling** | | |
| No education | 1,420 | 5.07 |
| 1–5 years | 1,339 | 4.79 |
| 6–10 years | 11,967 | 42.77 |
| 11 years and above | 13,256 | 47.37 |
| **Social groups** | | |
| SC | 6,570 | 23.48 |
| ST | 2,660 | 9.51 |
| OBC | 13,103 | 46.82 |
| Other | 5,650 | 20.19 |
| **Religion** | | |
| Hindu | 22,948 | 82.01 |
| Muslim | 3,638 | 13.00 |
| Christian | 601 | 2.15 |
| Others | 797 | 2.85 |
| **Wealth quintile** | | |
| Poorest | 4,616 | 16.49 |
| Poorer | 5,860 | 20.94 |
| Middle | 6,169 | 22.05 |
| Richer | 6,100 | 21.80 |
| Richest | 5,238 | 18.72 |
| **Exposure to mass media** | | |
| No exposure to mass media | 4,503 | 16.09 |
| Exposed to any one kind of mass media | 23,480 | 83.91 |
| **Discussed MH with CHW in last 3 months** | | |
| No | 27,490 | 98.24 |
| Yes | 493 | 1.76 |
| **Currently working** | | |
| No | 23,931 | 85.52 |
| Yes | 4,052 | 14.48 |
| **Takes bath during menstruation** | | |
| No | 913 | 3.26 |
| Yes | 27,070 | 96.74 |

**Table 1** (*continued*)

| Background characteristics | Frequency ($N = 27,983$) | Weighted percentage |
|---|---|---|
| **Consumption of alcohol** | | |
| No | 27,882 | 99.64 |
| Yes | 101 | 0.36 |
| **Place of residence** | | |
| Urban | 8,673 | 30.99 |
| Rural | 19,310 | 69.01 |
| **Region of residence** | | |
| North | 4,415 | 15.78 |
| Central | 7,941 | 28.38 |
| East | 6,093 | 21.78 |
| West | 3,435 | 12.27 |
| Southern | 5,347 | 19.11 |
| North-east | 752 | 2.69 |
| **Use of hygienic materials** | | |
| No | 13,341 | 47.68 |
| Yes | 14,642 | 52.32 |
| **Symptoms of RTI** | | |
| No | 26,865 | 96.00 |
| Yes | 1,118 | 4.00 |

**Notes.**

N, sample size; SC, scheduled caste; ST, scheduled tribe; OBC, other backward classes; MH, menstrual hygiene; CHW, community health workers; RTI, reproductive tract infections.

exclusive use of hygienic materials during menstruation, while approximately 4% reported experiencing symptoms of RTIs.

## Prevalence of RTIs by use of hygienic materials

The results of the bivariate analysis demonstrated a significant association between the utilization of hygienic materials during menstruation and the prevalence of reproductive tract infections (RTIs) among women. Specifically, only three out of 100 women (3.2% (95% CI [2.8–3.6])) who used hygienic materials during menstruation reported symptoms of RTIs. In contrast, about 5% (4.9% (95% CI [4.4–5.5])) of women who did not use hygienic materials during menstruation reported symptoms of RTIs (see Fig. 2).

## Spatial variation in the prevalence of RTIs and use of hygienic materials

Figure 3 reveals, the prevalence of RTIs was not homogeneously distributed across the Indian states. Among the larger states, highest prevalence of RTIs among women were observed in West Bengal (9.3%) followed by Himachal Pradesh, Rajasthan, Uttar Pradesh, and Gujarat (more than 4%). Among the smaller states, highest prevalence of RTIs was observed in Meghalaya, followed by Arunachal Pradesh, Sikkim, and Tripura. Across these above-mentioned states, the prevalence of RTIs was far more than national average of 4%. Among the larger Indian states, lowest prevalence of RTI was identified in Odisha (1.3%), followed by Uttarakhand and Maharashtra (less than 2%).

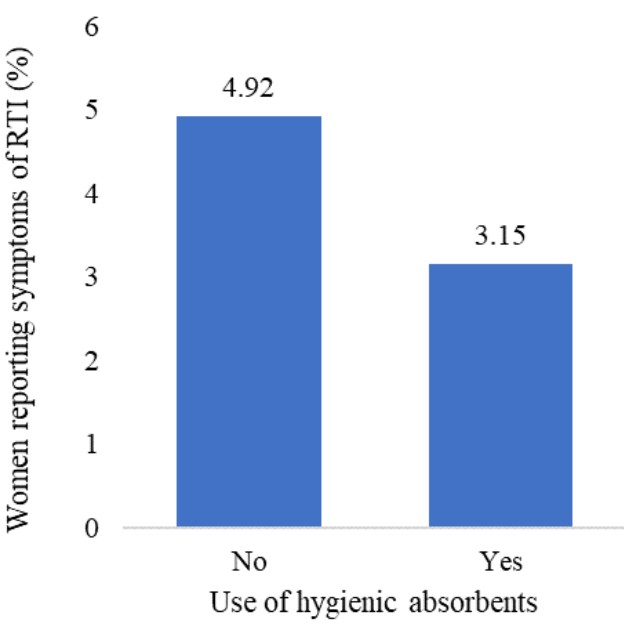

**Figure 2** Percentage of women reporting symptoms of reproductive tract infections by use of hygienic materials during menstruation, NFHS-5, 2019-21.

Among the UTs, the highest prevalence of RTIs among women was reported from Chandigarh (5.9%), followed by Delhi (3.7%). In contrast, the lowest prevalence of RTI was recorded in women from Puducherry (0.9%).

Figure 4 illustrates the spatial distribution of hygienic material use among women. Approximately 80% of women from Tamil Nadu, followed by Telangana and Haryana, reported use of hygienic materials during menstruation. Conversely, the lowest rates of hygienic material use were reported by women from Uttar Pradesh (30.8%), followed by Chhattisgarh (31.0%), Bihar (34.2%), and Madhya Pradesh (35.5%).

Among the UTs, Chandigarh (94.6%) exhibited the highest use of hygienic materials, closely followed by Andaman & Nicobar Islands (89.2%). In contrast, the lowest use of hygienic material was reported by women from Ladakh and Jammu & Kashmir (below 47%).

## Impact of hygienic material use on RTIs

The results presented in Table 2 provide estimates of the impact of hygienic material use during menstruation on RTIs among women in India.

Before matching, the initial estimates from the unmatched sample indicated that in the control group (women who did not use hygienic materials during menstruation), 48 per 1,000 women reported symptoms of RTIs. In contrast, in the treated group (women who used hygienic materials during menstruation), a lower proportion of women, specifically 31 per 1,000, reported symptoms of RTIs. These initial estimates from the unmatched sample suggest that there was a notable difference in the prevalence of RTIs between the

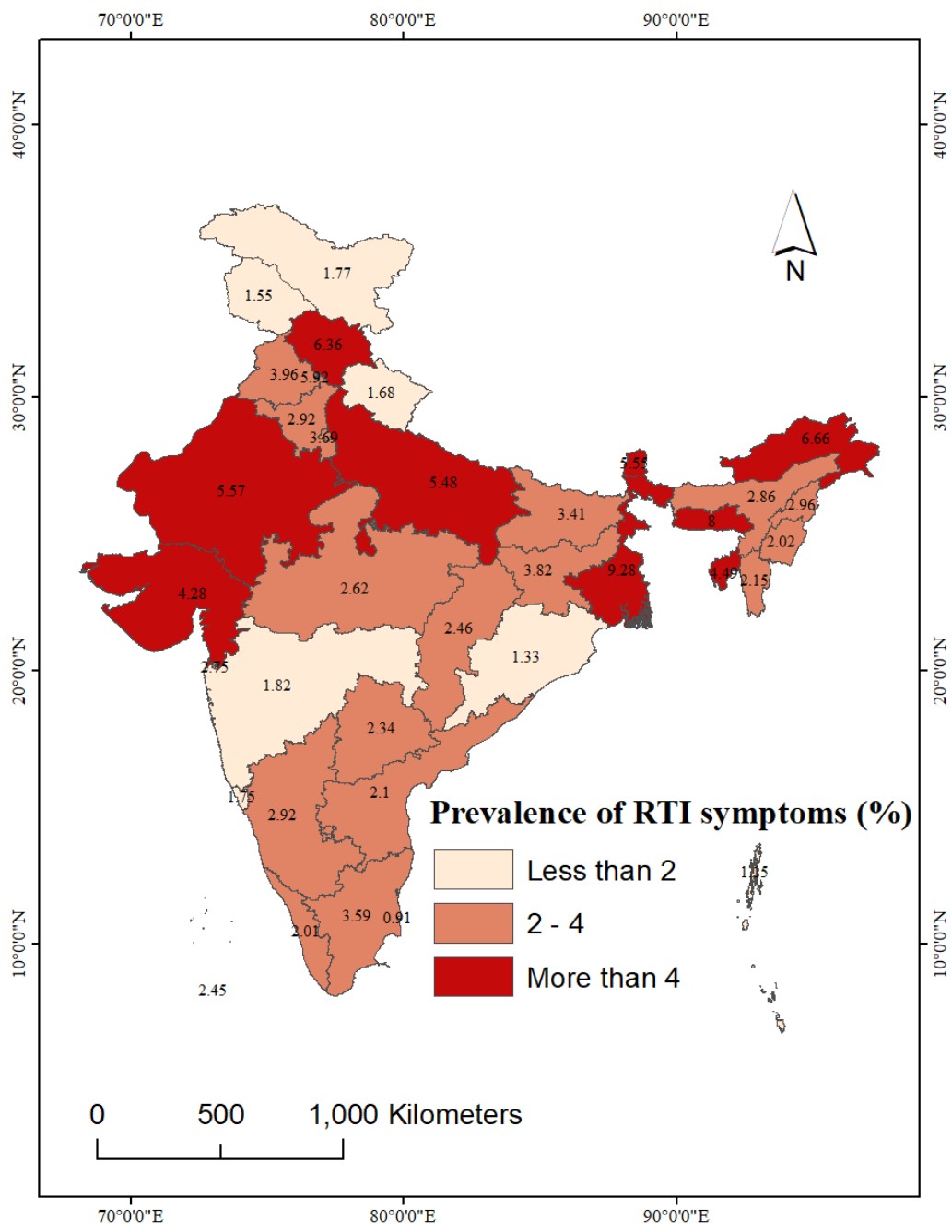

**Figure 3** State-wise prevalence of self-reported RTIs among women in India, NFHS-5, 2019-21. National and sub-national boundaries base layer source: https://spatialdata.dhsprogram.com/boundaries/#view=table{&}countryId=IA.

two groups of women. To provide a more accurate assessment of the impact of hygienic material use on RTI prevalence, we employed PSM.

After matching, the ATT estimates indicated that among women not using hygienic materials (control group), approximately 41 per 1,000 women reported symptoms of RTIs

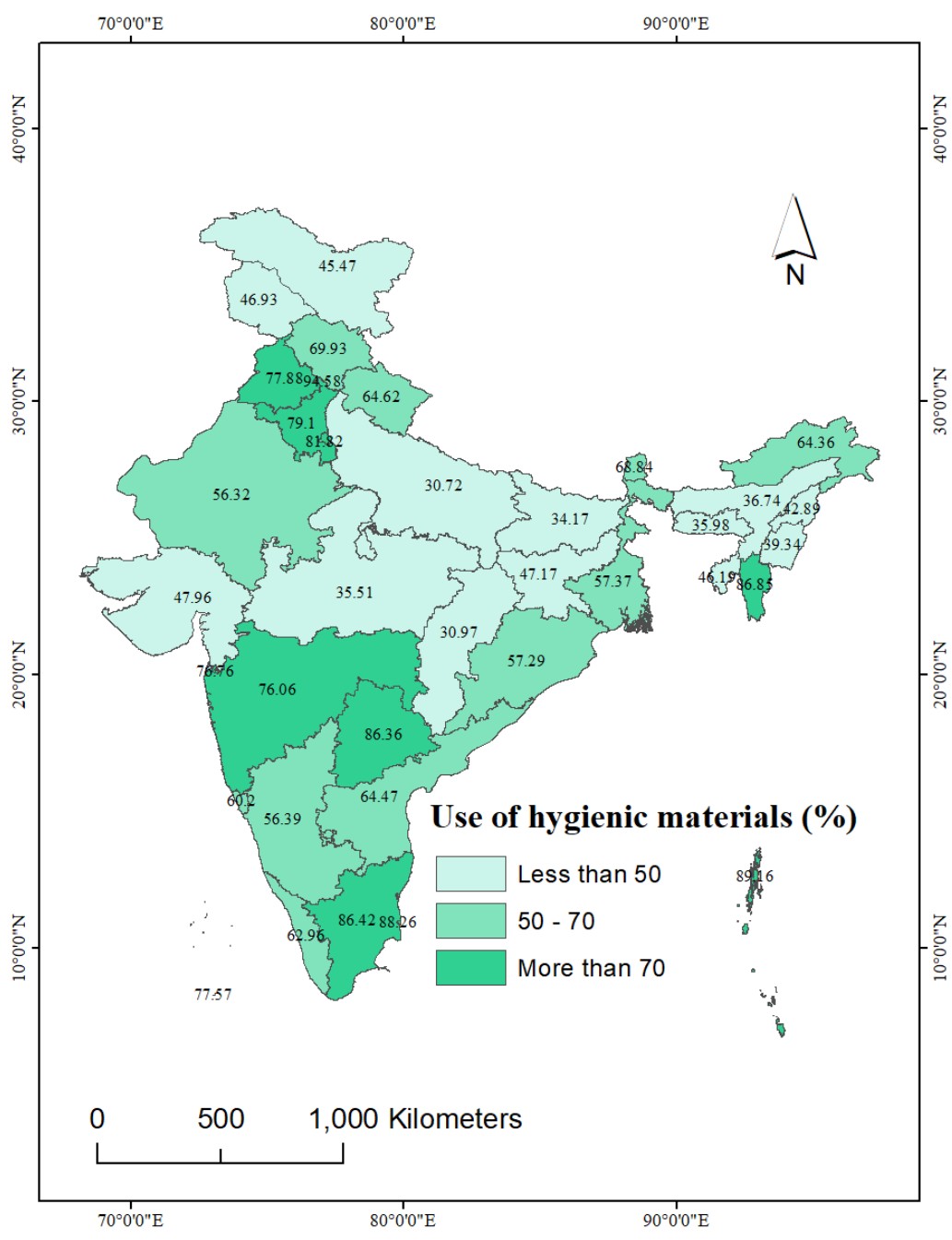

**Figure 4** State-wise use of hygienic materials during menstruation among women in India, NFHS-5, 2019-21. National and sub-national boundaries base layer source: https://spatialdata.dhsprogram.com/boundaries/#view=table{&}countryId=IA.

(ATT: 0.0416). In contrast, among the women using hygienic materials (treated group), 31 per 1,000 women reported symptoms of RTIs (ATT: 0.0315). These ATT estimates suggested that even after the matching process, there remained a significant difference of 10 per 1,000 RTI cases between women who used hygienic materials and those who did

**Table 2 Estimates of the impact of use of hygienic materials during menstruation on RTI among women in India, NFHS-5, 2019-21.**

| Impact of use of hygienic materials on RTI | Treated | Controls | Difference | S.E. | T stat | P value |
|---|---|---|---|---|---|---|
| Unmatched | 0.0315 | 0.0484 | −0.0169 | 0.0023 | −7.21 | <0.001 |
| ATT | 0.0315 | 0.0416 | −0.0101 | 0.0040 | −2.55 | <0.050 |
| ATU | 0.0484 | 0.0448 | −0.0036 | . | . | |
| ATE | | | −0.0068 | . | . | |

**Notes.**

RTI, reproductive tract infections; ATT, average treatment effect on treated; ATU, average treatment effect on untreated; ATE, average treatment effect on untreated; S.E., standard error; treated, women who used hygienic materials during menstruation; control, women who did not use hygienic materials during menstruation.

**Table 3 Description of the sample used in the matching analysis.**

| Treatment assignment | Sample size | | Total |
|---|---|---|---|
| | Off support | On support | |
| Untreated | 21 | 13,935 | 13,956 |
| Treated | 14 | 14,013 | 14,027 |
| Total | 35 | 27,948 | 27,983 |

not. This implies that the use of hygienic materials is associated with a lower prevalence of RTIs among women.

## Verification of estimates
### Common support
Examining the overlap and common support region between the treatment and comparison groups is a crucial step in the study, as the computation of the average treatment impact (ATT) relies solely on the observations within the common support area. Table 3 reveals that among the total sample of 27,983 women, 35 observations were excluded. Specifically, 14 observations were removed from the untreated and 21 from the treated groups due to the unavailability of suitable matches, leaving a remaining sample size of 23,948 women.

### Quality of matching
Figure 5 displays the distributions of propensity scores between two groups of women: those who reported using hygienic materials during menstruation (treated group) and those who did not (control group). In this figure, control group women's propensity scores are illustrated as bars below a vertical line, whereas treated women's scores are presented as bars above the line. Through propensity score matching, it is observed that the distributions of propensity scores in the treatment and control groups become comparable, as depicted in the figure. This concordance suggests that the characteristics of the treated and untreated women were strikingly similar.

### Balancing test
Figure 6 indicates the standardized % bias of each matching variable was dissimilar prior to matching. Notably, the standardized % bias was observed to be most pronounced in the
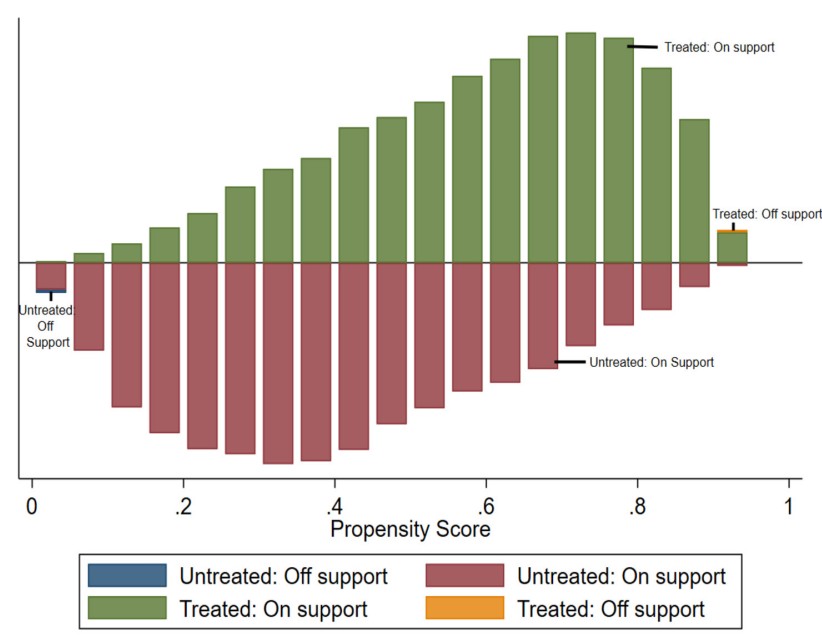

**Figure 5 Common support for use of hygienic materials.**

variables of respondents' household wealth status, years of schooling, place of residence, and mass media exposure. However, after matching the differences in standardized % bias of nearly all covariates were sufficiently reduced. For instance, after the matching the % bias was reduced (see % reduction bias) by more than 80% in almost each matching variable (see Appendix S3). Also, after the matching, the % bias reduced to less than 10% for all variables, which indicates a good matching (*Singh, 2016*). Furthermore, after the matching procedure, the *p*-values resulting from t-tests mostly exceeded 0.05, suggesting the absence of statistically significant differences between the groups. However, two variables—age and wealth factor score—still exhibited *p*-values >0.05, indicating statistically significant differences between the groups.

Moreover, after matching, the variance ratios for each matching variable approached a value of 1, including age and wealth factor score. This signifies a favourable balance between the control and treated groups (*Zhang et al., 2019*) (see variance ratios in Appendix 3). In simpler terms, it implies that our matching process successfully brought the groups into closer alignment, even for variables that initially showed substantial differences.

### Significance of the model

Table 4 provides a comprehensive assessment of the model's significance. The pseudo-$R^2$ reduced significantly from 0.156 in unmatched sample to 0.001 in matched sample indicating, there were no systematic differences in the distribution of covariates between treated and control groups. Furthermore, the mean bias and median bias also reduced to 1.6 and 1.2 respectively, far below the limit (>20) recommended by

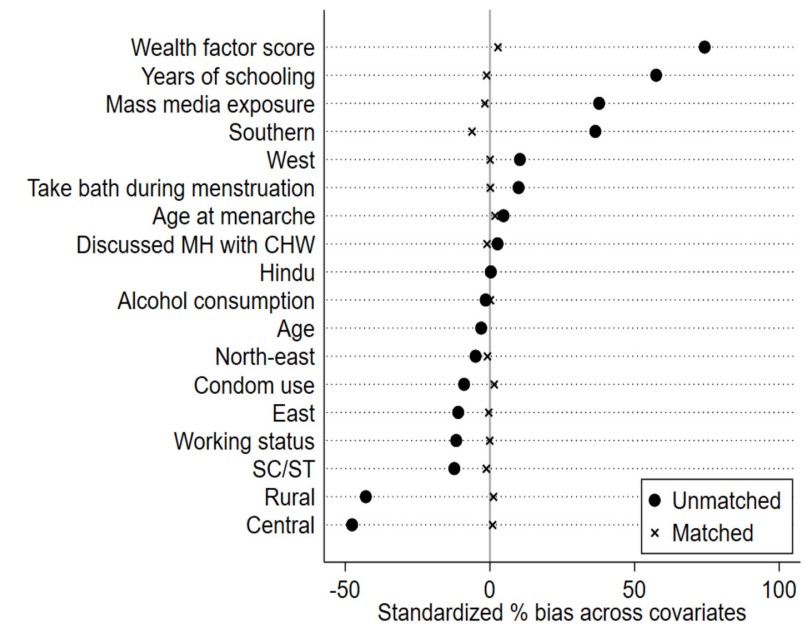

**Figure 6    Reduction in standardized bias before and after matching.**

**Table 4    Overall significance of model after matching.**

| Sample | Pseudo R2 | Likelihood Ratio chi2 | $p > $ Chi2 | Mean Bias | Med Bias | Rubin's B | Rubin's R | % variance |
|---|---|---|---|---|---|---|---|---|
| Unmatched | 0.156 | 6061.6 | <0.001 | 21.0 | 10.7 | 99.2 | 0.8 | 60 |
| Matched | 0.001 | 50.1 | 0.100 | 1.6 | 1.2 | 8.5 | 1.1 | 20 |

*Rosenbaum & Rubin (1985)*. The same table also shows that the Rubin's B value in the matched sample was reduced to 8.5, way below the threshold limit of 25. Also, Rubin's R which should fall between 0.5 and 2.0, turned out to be 1.1.

### Sensitivity analysis

The findings of the sensitivity analysis are presented in Table 5. Γ or Gamma represents the odds of differential assignment due to unobserved factors. p_mh+ and p_mh − stand for the upper and lower boundaries of the significance levels, respectively. When the value of Γ is 1, it signifies the absence of any hidden bias in the model.

   In our study, when Gamma or Γ = 1, the p_mh+ and p_mh − both were 0.001, signifies strong evidence that the use of hygienic materials causes RTIs (p_mh+ and p_mh- 0.001). When the p_mh+ reaches 0.209, when the bias introduced by potential confounders increased to 1.2 (Γ = 1.2), which means that when the positive selection bias introduced in our model is 1.2 or higher, our model ends with a biased estimation of the treatment effect of use of hygienic materials.

**Table 5  Sensitivity analysis.**

| Gamma | Q_mh+ | Q_mh- | p_mh+ | p_mh- |
|---|---|---|---|---|
| 1 | 3.17 | 3.17 | 0.001 | 0.001 |
| 1.1 | 1.93 | 4.41 | 0.027 | <0.001 |
| 1.2 | 0.81 | 5.55 | 0.209 | <0.001 |
| 1.3 | 0.14 | 6.60 | 0.443 | <0.001 |
| 1.4 | 1.10 | 7.59 | 0.136 | <0.001 |
| 1.5 | 1.99 | 8.52 | 0.023 | <0.001 |
| 1.6 | 2.82 | 9.39 | 0.002 | <0.001 |
| 1.7 | 3.61 | 10.22 | <0.001 | <0.001 |
| 1.8 | 4.35 | 11.01 | <0.001 | <0.001 |
| 1.9 | 5.06 | 11.77 | <0.001 | <0.001 |
| 2 | 5.73 | 12.49 | <0.001 | <0.001 |

**Notes.**

Gamma, odds of differential assignment due to unobserved factors; Q_mh+, Mantel-Haenszel statistic (assumption: overestimation of treatment effect); Q_mh-, Mantel-Haenszel statistic (assumption: underestimation of treatment effect); p_mh+, significance level (assumption: overestimation of treatment effect); p_mh-, significance level (assumption: underestimation of treatment effect).

## DISCUSSION

The findings of our study revealed a concerning prevalence of self-reported RTIs among women in India, with four out of 100 women reporting such infections. However, we also identified that the prevalence of RTIs among women varied widely across the states and UTs of India. Additionally, the study utilized the PSM approach, a quasi-experimental method, to evaluate the impact of hygienic material use on RTIs among women in India. Using this method, we identified that women who used hygienic materials during menstruation had lower chance of RTIs compared to women who did not use hygienic materials during menstruation. This finding is consistent with other published works that have also found a link between use of hygienic materials during menstruation and RTIs (*Anand, Singh & Unisa, 2015*; *Das et al., 2015*; *Das et al., 2021*; *Torondel et al., 2018*; *Vishwakarma, Puri & Sharma, 2020*; *Sommer et al., 2021*). The use of hygienic materials ensures the absorption of menstrual blood, which reduces the chances of leakage and staining (*Anand, Singh & Unisa, 2015*). These materials keep the vaginal area clean and dry, which further reduces the risk of RTIs (*Almeida-Velasco & Sivakami, 2019*).

Previous studies have noted that unhygienic menstrual practices can lead to genital infections, RTIs or Urinary Tract Infections (*Anand, Singh & Unisa, 2015*; *Almeida-Velasco & Sivakami, 2019*). The mechanism behind this relationship is likely multifactorial. For instance, a study conducted in a hospital setting in India reported that women who did not change their menstrual cloths frequently (less than twice a day) and wore them for extended durations were at an elevated risk of RTIs (*Torondel et al., 2018*). This prolonged use of cloths can lead to the accumulation of menstrual blood into the vagina, potentially disrupting the vaginal ecosystem and promoting the development of Bacterial Vaginosis (BV), a common type of RTI, the most common type of RTIs (*Phillips-Howard et al., 2016*; *Torondel et al., 2018*). This prolonged use of cloths can lead to the accumulation of

menstrual blood in the vaginal canal, potentially disrupting the vaginal ecosystem and promoting the development of Bacterial Vaginosis (BV), a common type of RTI (*Torondel et al., 2018*). This condition often results into vulvo-vaginal candidiasis (VVC), another prevalent RTIs among women (*Torondel et al., 2018*). Once menstrual cloths come into contact with Candida, their removal without adequate washing and drying becomes challenging (*Torondel et al., 2018*). Additionally, several studies have documented that cloth and reusable materials require longer drying times, prompting many women to wear and even store them while damp (*Das et al., 2015*; *Das et al., 2021*). This perpetuates a moist environment, further increasing the risk of VVC among women (*Das et al., 2021*). These factors may contribute to higher prevalence of RTIs among women who use unhygienic materials, including cloths, during menstruation.

It is worth noting that some of the recent studies have argued that, under certain conditions, cloths can also be considered hygienic if they are appropriately washed, dried, and stored for future use (*Hennegan & Montgomery, 2016*; *Almeida-Velasco & Sivakami, 2019*). Unfortunately, in low-middle income countries like India, due to a lack of access to water, sanitation and hygiene facilities, proper cleaning and drying cloths can be challenging (*Van Eijk et al., 2016*). Additionally, washed menstrual cloths are often dried in cold and dark locations or hidden beneath other garments, due to the taboos and misconceptions surrounding menstruation (*Sharma et al., 2017*). As a result, women often compromise their MHH, ultimately resulting into getting symptoms of RTIs (*Anand, Singh & Unisa, 2015*; *Van Eijk et al., 2016*). Therefore, the government should take proactive steps to raise awareness and provide training to women and girls on the proper management of cloth materials if they are chosen as a menstrual hygiene option.

Acknowledging the substantial impact of menstrual hygiene on women's health and well-being, including susceptibility to RTIs, the Government of India has taken significant steps to promote the adoption of hygienic materials during menstruation, (*NDTV, 2019*; *Ram et al., 2020*; *Ministry of Jal Shakti, 2022*). Central and state-specific initiatives and policies have been put in place to provide support in the form of subsidized and free hygienic materials (*Gupta*; *Shah, 2016*; *Sarkar, 2020*; *Karan, 2021*; India *Today, 2021*). While some of these initiatives have achieved success, others have encountered challenges due to various obstacles, leading to an uneven adoption of hygienic materials across different regions and social strata in India. In addition to these efforts, it is crucial that the government ensures that women have a range of informed choices when selecting menstrual products, taking into consideration affordability and individual needs. These multifaceted interventions hold the potential to reduce the incidence of RTIs among women, significantly.

Although this study examined the spatial variation in the prevalence of RTIs and measured the impact of hygienic material use during menstruation on the prevalence of RTIs among women in India, it has certain limitations that should be noted here. Firstly, the utilization of self-reported data in this study may have introduced potential reporting bias. This is because societal stigmas and cultural taboos pertaining to sexual and reproductive health could have led to underreporting of RTI symptoms by women. Secondly, due to data constraints, our analysis excluded women aged 25–49, as questions regarding menstrual

hygiene were not posed to this age group in the NFHS-5 survey. Also, the NFHS-5, did not collect information on the frequency or regularity of hygienic material use among women, which limit our ability to comprehensively investigate the relationship between frequency of hygienic material use and RTI prevalence among women. Additionally, the study did not investigate the specific types of RTIs that women in India are most susceptible to, and whether the use of hygienic materials is equally effective in reducing the risk of all types of RTIs. Further research is needed to identify the specific types of RTIs that are most prevalent among women in India and to evaluate the effectiveness of hygienic materials in reducing the risk of these RTIs. Another limitation of this study is its exclusive focus on women due to data constraints. Consequently, we were unable to include information on menstruating trans and non-binary individuals since the NFHS-5 dataset did not cover this demographic. This limitation means that the study's findings may not offer a complete representation of the diverse population affected by the subject under investigation, potentially overlooking their unique experiences and health implications (*Babbar et al., 2023*). Future research should strive to include a broader spectrum of individuals to provide a more comprehensive understanding of this issue.

## CONCLUSION

In conclusion, our study has unveiled compelling evidence that the exclusive use of hygienic materials during menstruation is associated with a reduced prevalence of RTIs among women in India. These results underscore the pressing need for tailored interventions in states and UTs with elevated RTIs prevalence and highlight the potential role of promoting hygienic material use in mitigating the burden of RTIs among women in India.

### Funding

This research was supported by the Banaras Hindu University's Institute of Eminence (IOE) Seed Grant awarded to Dr. Aditya Singh (Grant No. R/Dev/D/IoE/Equipment/Seed Grant II/2022-23/48726), while Mahashweta Chakrabarty received support in the form of a Junior Research Fellowship from the University Grants Commission, India (Reference Number: 200510082749). The funders had no role in study design, data collection and analysis, decision to publish, or preparation of the manuscript.

### Grant Disclosures

The following grant information was disclosed by the authors:
Banaras Hindu University's Institute of Eminence (IOE): R/Dev/D/IoE/Equipment/Seed Grant II/2022-23/48726.
University Grants Commission, India: 200510082749.

### Competing Interests

Aditya Singh is an Academic Editor for PeerJ.

## Author Contributions

- Mahashweta Chakrabarty conceived and designed the experiments, performed the experiments, analyzed the data, prepared figures and/or tables, authored or reviewed drafts of the article, and approved the final draft.
- Aditya Singh conceived and designed the experiments, performed the experiments, analyzed the data, prepared figures and/or tables, authored or reviewed drafts of the article, and approved the final draft.

## Data Availability

The dataset used in this study is publicly available at The DHS Program: https://dhsprogram.com/data/available-datasets.cfm.

## Supplemental Information

Supplemental information for this article can be found online at http://dx.doi.org/10.7717/peerj.16430#supplemental-information.

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
