# Peer review of "Assessing the link between hygienic material use during menstruation and self-reported reproductive tract infections among women in India: a propensity score matching approach"

_PeerJ, doi:10.7717/peerj.16430_

## Round 0.1 · original submission · Major Revisions

Many thanks for submitting this paper. It has potential and could be considered for publication if comments are considered, reviewed and integrated. In addition to the comments by the reviewers, here are two more general comments:

1. Better justify using a nearest neighbour matching compared to other matching techniques.
2. Sensitivity analysis is an essential part of propensity score matching. Consider adding this.

Reviewer 1 ·

Basic reporting

1. Authors should be consistent with the introduction to results. Some dissimilarities can be seen throughout the manuscript.
2. The introduction is quite long. We recommend to shorten it up to 1.5 pages
3. They should write briefly in the methodology.
4.The discussion is quite long. We recommend to shorten it
5. References should be reviewed. We recommend to use references from recent studies (less the 5 years)

Experimental design

1.Ethical statement to the secondary data use should be available in this study
2.A dataset and a do-file code have to be available. Therefore, we strongly recommend authors to use a github link.
3. Data processing technique used have to described in this study.
#Treatment variable
‘’The question contained seven response categories, which were classified into two groups: i) sanitary napkins, ii) locally made napkins, iii) tampons, and iv) menstrual cups were categorized as 'hygienic methods', while v) cloth, vi) nothing, and vii) others were categorized as 'unhygienic methods'. Based on the responses, a binary outcome variable was created, whereby women were categorized as "exclusive users of hygienic methods" (coded as "1") if they used only sanitary napkins, locally made napkins, tampons, and/or menstrual cups. The remaining women who used cloth, other materials, nothing, or a combination of both hygienic and unhygienic methods were labelled as "users of unhygienic methods" (coded as "0") (Singh et al., 2022b, c).”
This categorization should make some confusions: There was some women using both hygienic and unhygienic materials.Evenif they use both two kind of material, you don’t know if hygienic material were not more than unhygienic materials.
-Moreover, the frequency of exclusively hygienic material use should be stated. If it was not used to each menstruation period, reproductive tractus infections could occurred during this period.
-Last but not least, women’s underlying and living conditions for could be taken account in the RTIs occurrence.
#Propensity Score Matching (PSM)
1. Authors stated the reseason to choose the statistical analysis method (PSM), which is good. They also to described method. However, we recommend to add an reference (from World Bank /DME and / or Rosenbaum P& Rubin D.
2. PSM model, a probit/logit model, was used with the equation. We recommend to rewrite it as mathematical equation
3. We recommend to authors to specify (with a reference) the matching technique used.
Kernel matching is preferred to one-on-one matching as it offers better matching controls
4. Authors should state the sufficiency or insufficiency of the sample size post matching. If insufficient, we recommend to apply consequently adequate methods.
5. Statistical significance was not stated.
Balancing test
1. Authors assessed the quality of post matching balance. We recommend to use also the variance ratios and percentage of bias.

Validity of the findings

1.Results should be more oriented to the main objective of the study.
2. The discussion is quite long. It could be shortened and be more oriented the study objectives.
3. Discussion and introductions present dissimilarities:
L106-L108: “Therefore, the present study attempts to evaluate and measure the impact of using hygienic materials during menstruation on the reduced prevalence of self-reported RTIs among young 108 Indian women using propensity score matching.
L374-L375: “Although this study provides important insights into the prevalence and risk factors associated with RTI among young women in India, it has certain limitations”.
We recommend authors to review the two sections

Reviewer 2 ·

Basic reporting

Thank you for writing this paper. The paper's research question is valuable, and I enjoyed reading it. I am suggesting a major revision of your paper before it can be considered publishable for a wider audience of the journal. I am not able to comment of the PSM design as it is beyond my area of expertise.

1. The literature on menstruation has moved from menstrual hygiene management to menstrual health and hygiene. Please make the corrections in the manuscript.

2. Why does the study focuses on only 28 states? Where are the remaining 8 UTs? Is this the reason for the lower sample size of 27k, instead of 36k? This is the major flaw of the paper and needs a very strong justification from the authors.

3.  You mention, "The study found significant regionalvariations in RTI prevalence, with the highest rates observed in West Bengal, Rajasthan, Uttar Pradesh,and Gujarat. However, the burden of RTI was found to be lower in states where women reported higheruse of hygienic materials during menstruation, such as Maharashtra, Andhra Pradesh, and Telangana."Shouldn't there be other state-level issues that one needs to account for before making such a statement? Did you control for the state FE? See Babbar, K., Rustagi, N., & Dev, P. (2022). How COVID‐19 lockdown has impacted the sanitary pads distribution among adolescent girls and women in India. Journal of Social Issues.

4. In the conclusion portion of the abstract you write, "Improving access to hygienic materials and menstrual hygiene facilities, as well asincreasing awareness and education around menstrual hygiene, could help to reduce the burden of RTIsin high-prevalence states."The conclusion looks too stretchy to include awareness and education when these are not the direct mechanism that you've examined in the paper.

5. You use multiple terms in the paper menstrual hygiene management, menstrual hygiene practices, and so on. Please read the paper carefully and use one consistent term.

6. There are multiple grammatical errors in the manuscript. Please get a proof check done before sending the revised manuscript.

Experimental design

7. I am a little confused with the construction of the dependent variable RTI. See Bhasin 2020.
They include the following as a part of their DV RTI: "state-level modules included three questions about RTIs for women who reported being sexually active, irrespective of their marital status: experience of ailments due to sexual contact; bad-smelling abnormal genital discharge; and the presence of genital sores or ulcers, all within the twelve- month period before the survey." (Bhasin et al., 2020)
Bhasin, S., Shukla, A. & Desai, S. Services for women’s sexual and reproductive health in India: an analysis of treatment-seeking for symptoms of reproductive tract infections in a nationally representative survey. BMC Women's Health 20, 156 (2020). https://doi.org/10.1186/s12905-020-01024-3

8. A detailed discussion of the control variables needs to be given. Looking at the list, I am still wondering how the consumption of alcohol is affecting the RTIs.

9. Also, if I am not wrong, then, variables like the discussion of menstrual hygiene with community health workers and working status are not available for all the 27k women and hence, reduces the overall sample size. It might be good to add them in the robustness checks and not in the final analysis.

Validity of the findings

10. The argument for using PSM needs to be sharpened further. 

11. Line 172 needs a citation. 

12. Line 184 needs a better equation. Please look into a few PSM papers published in the economics journals and make the changes for comments 9-11.

13. Line 185-186 Needs citation again.

14. Based on Table 1, justify the usage of the variables given that all of them have less than 5% variation.1) Discussed MH with CHW in last 3 months2) Currently working3) Consumption of alcohol

15 Line 234-235 says only 3/10 women who use period products (PP) experience RTI. 30% is still a huge number. Do you wanna double check these numbers?

Line 236-238. You say 5% of women who do not use PP experience RTI. So it is better not to use PP? Isnt it?

16) I am still confused with the final numbers reported:

In the final report released by IIPS, Bihar had the lowest number, and now since the sample reduces due to limited info, Bihar doesn't even turn up as one of the low-performing states. I am a  little surprised with these findings and I would suggest a re-check on these numbers. Also, see https://thewire.in/health/what-nfhs-5-data-tells-us-about-indian-womens-use-of-period-products It has a detailed descriptive analysis of PP usage in India using NFHS-5 data.

17) Line 275-277: Add in a footnote why 21 observations were excluded?

Additional comments

18 I fail to understand the need to conduct a detailed review of the health policies linked to menstruation from lines 334 to 373.
19 The paper also needs a section on the author's discussion of their findings, not just matching their results with the existing research or policy decisions. This would help reflect the authors' understanding of the topic being discussed.
20 You can further add a limitation that data was linked to only women and does not include data on trans and non-binary individuals who menstruate.
Babbar, K., Martin, J., Varanasi, P., & Avendaño, I. (2023). Inclusion means everyone: standing up for transgender and non-binary individuals who menstruate worldwide. The Lancet Regional Health-Southeast Asia.

---

## Round 0.2 · Minor Revisions

1. The authors mention the use of nearest neighbour and indicate it was based on the recommendations from Rosenbaum and Rubin (1985) and Austin (2011) but fail to state what about this matching technique that makes it better compared to other matching techniques. For context, there are a number of techniques; Kernel, Mahalanobis, radius, etc. It will be good for authors to indicate the added value of this matching technique.

2. Significance of the model: The statement “After matching there should be no systematic differences in the distribution of covariates between treated and control groups and therefore, the pseudo-R2 should be fairly low” is a direct copy and paste from the Caliendo and Kopeinig. If this is so, it is advised authors put that in inverted commas. If not, it is recommended to rephrase

3. Post matching estimates: After propensity score matching, there are some variables that appear to still be statistically significant different (such as age, wealth factor). While this is not necessarily an issue in the context of propensity score; the propensity scores are calculated using these background characteristics and a region of common support established, it will be good to comment on this in the results or discussion section.

4. To better visualize the results of the balancing diagnostics, it will be good to develop a graph. In STATA, adding the “,both graph” to the end of the “pstest” command should work.

5. Comment 9 of the second reviewer. Although the justification offered by the authors regarding the inclusion of alcohol use is valid, the first paper cited (Froehle et al. 2021) is incorrect and does not support your argument. Just an FYI

---

## Round 0.3 · accepted · Accept

The manuscript has been greatly improved and warrants publication. Congratulations!